# The Number and Regional Distribution of Chinese Monks after the Mid-Qing Dynasty

Xuesong Zhang

Institute for the Study of Buddhism and Religious Theory, Renmin University of China, Beijing 100872, China; xuessong@hotmail.com

**Abstract:** The total number of ordination certificates issued between 1736 and 1739 was 340,112. Analyzing the amount and regional distribution of ordination certificates during the early Qianlong period is helpful for us in clarifying the amount and regional distribution of Chinese monks since the mid-Qing Dynasty. The total number of Buddhist monks did not change measurably during the two hundred years from Qianlong's reign until the Republic period, remaining between 600,000 and 700,000. Although the census in the 1930s did not cover Taoist monks, as previously discussed, their number may have been similar to that during Qianlong's reign. As a result, the number of monks (both Buddhist and Taoist) did not changed much after the mid-Qing Dynasty, despite many historical changes since the 19th century, such as population growth, the Taiping Heavenly Kingdom Movement, the promotion of education with temple property, and the warlord conflicts. The number of Buddhist monks in Northern China declined significantly from 1742 to 1936, while that in the regions along the midstream and downstream of the Yangtze River and in Southwestern China, it increased significantly. However, the geographical layout of Chinese Buddhism did not changed much, as there was neither a noticeable decline nor a noticeable revival in the number of monks and nuns.

**Keywords:** Buddhist monks; Buddhist geography; ordination certificate

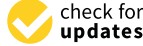



## Preface

At the beginning of the study of modern Buddhist history, some Christian publications, such as *The Chinese Recorder*, and missionaries, such as Karl Ludvig Reichelt (艾香德, 1877–1952), began paying great attention to what they called the "revival" of Buddhism in China. After the widespread use of Christian missionaries, the modern "revival" of Buddhism became a well-known concept. In particular, the "revival" paradigm of modern Chinese Buddhism put forward by Holmes Welch has had a great influence in academic circles (Welch 1968). Although some scholars thought that the innovation of modern Buddhism in China failed in the 1960s and 1970s,[1] the "revival" paradigm has always been in a dominant position, which has influenced the research of mainland scholars after 1979.[2] In the new century, with the deepening and refinement of research, the general works on the history of modern Buddhism in China have gradually decreased, and the focus of research has shifted from whether or how modern Buddhism itself was "revived".[3]

Many people debate the decline of Buddhism in modern China and believe that there are various causes for this decline. However, the reasons they have found are often contradictory. On the one hand, some of them think that the decline in the number of monks in China (with some even claiming that the number of monks in China dropped from one million to ten thousand) is a sign Buddhism's decline. On the other hand, others believe that the abolition of the system of the ordination certificate is responsible for the decline of Buddhism in China. Without this system, the number of monks was out of control, and there were too many monks, which attracted a large number of social idlers into Buddhism.

The sharp increase in the number of monks, however, caused a serious decline in the quality of monks, and the original temple economy could not afford these new monks, leading to further deterioration of the temple economy.

However, the number of monastics and Buddhist monasteries in a particular place and period is the most significant indication for gauging the evolution of Chinese Buddhism. The mid-18th century repeal by the Qing dynasty of the ordination certificates, which had been in effect for more than a millennium, was a pivotal moment in the evolution of the modern Buddhist institution. The introduction of ordination certificates was perhaps one of the most significant means by which the government administered monasticism during the entire time of imperial China. It reflected the subjugation of ecclesiastical power to the monarchy and brought the monastic community under effective secular rule.

The complete abolishment of the ordination certificate (度牒) during Qianlong's reign, which had lasted for over one thousand years, was a milestone in the administration of the Chinese government towards monks. Emperor Qianlong tried to reinforce the administration of the ordination certificate before he abolished it. In the year 1738, in one of his imperial edicts, Emperor Qianlong proclaimed the aim of issuing ordination certificates to monks: "Just as the Tithing System (保甲), ordination certificates can verify the legitimacy of a monk, in this way, no one can pretend to be a monk" (令有所稽考，亦如民間之有保甲，不至藏奸。 *Daqing huidian shili*, 大清會典實例, V501).

According to the traditional view, the abolishment of the ordination certificate was directly related to the policy of abolishing the head tax in the mid-Qing Dynasty. However, as pointed out by Mr. Yang Jian (楊健) in his new book *Qing wangchao fojiao shiwu guanli* (清王朝佛教事務管理, Administration of Buddhist Affairs in the Qing Dynasty), "the ordination certificate is related to the head tax, as it has the economic function of exempting the head tax, but this economic function is secondary and subsidiary. The ordination certificate is the product of the contest between the feudal dynasties and Buddhism, the manifestation of the relationship between kingship and Buddhism, and one of the typical symbols of the Sinicization of Buddhism. The most fundamental thing about an ordination certificate is the relationship between politics and religion, which is the first and foremost thing. The economic relationship embodied in the ordination certificate is peripheral. It can be assumed that, if the Qing Dynasty could still effectively manage the monks and nuns through the ordination certificate, that is, even if there was 'a reform that abolished the head tax and merged it into the agricultural tax' (攤丁入畝), the ordination certificate system may not be abolished because the most fundamental function of the ordination certificate to manage the monks and nuns has not disappeared. In other words, abolishing the head tax and merging it into the agricultural tax is not a sufficient condition for the abolition of the system of the ordination certificate" (J. J. Yang 2008). Mr. Yang Jian believes that, apart from the causes such as "a reform that abolished the head tax and merged it into the agricultural tax" and losing the economic function of the ordination certificate, the external reasons for the abolishment of the ordination certificate are two: one is that the administrative system of Buddhist affairs in the Qing Dynasty was basically determined, and the other is the perfunctory actions of local officials. In my opinion, the main principle that Mr. Yang has stated is very reasonable. Regardless of whether there is an ordination certificate or not, unless it is specially approved, the Qing Dynasty levied taxes on monasteries and estates, but there is still room for further discussion on the analysis of the specific reasons for the abolition of ordination certificates.

Mr. Yang Jian believes that during Qianlong's reign, the monk official system had matured and could fully utilize role of 'ruling monks by monks'. Additionally, the secular regime had extensively intervened in Buddhist affairs through the Tithing System, enabling the Qing Dynasty to manage monks and nuns at the most basic level of society. Finally, after more than a century of efforts, the *Daqing Lvli* (大清律例, The Case Summaries of Laws of the Great Qing Dynasty) had been formulated and served as the core of all laws, and the legal system of the Qing Dynasty was fully established by 1740. Mr. Yang Jian be-

lieves that the rulers of the Qing Dynasty did not worry about any adverse consequences following the abolition of the ordination certificate system" (J. Yang 2008, p. 157).

The points raised by Mr. Yang Jian, such as the mature monk-official system, the dynasty's intervention in Buddhism with the Tithing System, and the consummate legal system, were not unique to the Qing dynasty. These are all stories from the previous dynasties, yet they did not abolish the ordination certificate. Moreover, current academic research on the history of the legal system depends not only on the formulation of rules and regulations but also on their implementation, feasibility, and to what extent they can be executed. Beased on historical data, it is challenging to assert that "the Qing Dynasty could implement the administration of monks and nuns to the most basic level of society"[4] Additionally, if these statements were true and the management of monks and nuns were fully implemented in the Qing Dynasty, there would not have been any "perfunctory actions of local officials" that Mr. Yang also noticed.

As a matter of fact, the abolition of the ordination certificate system was not an active choice of the Qing Dynasty, but rather a helpless one. It was the result of the complete failure and collapse of the ordination certificate system and its collapse. To some extent, it can be said that the religious policy of the Qing Dynasty failed. Nonetheless, the number and geographical distribution of the last ordination certificate issued by the Qing Dynasty are still useful for studying the overall situation of monks in China during that period.

The total number of ordination certificates issued between 1736 and 1739 was 340,112. Analyzing of the quantity and geographical distribution of the first few years of Qianlong's reign, and comparing it with the investigation and statistics of Buddhism in the Republic of China, will help us understand the amount and regional distribution of Chinese monks since the mid-Qing Dynasty and even later.

Using geographic information systems (GIS) and data statistics to study the development and evolution of Buddhism can provide a rough quantitative evaluation of religion. Although this method has limitations due to the data itself, it can objectively reflect the hidden rules of data compared to speculation. Scholars have started to use GIS and data statistics to study the geographical distribution of monks in China. For instance, Yang Fenggang edited an atlas that used GIS and data statistics to produce more than 150 full-color maps, including six case studies analyzing the distribution of major religions in China at the national, provincial, and county levels, describing the main organizations, beliefs, and ceremonies of major religions in China, as well as the social and demographic characteristics of their followers (F. Yang 2018). Zhong Yexi et al. used religious site data in China and analyzed the temporal and spatial changes of the distribution of religious sites in China, taking the city as the research unit and Buddhism, Taoism, Islam, and Christianity as the research objects. They applied the methods of statistics and spatial analysis to explore the distribution and development trends of major religions in China (Zhong and Bao 2014). Huanyang Zhao et al. investigated the spread processes of Buddhism, Taoism, and Christianity in the coastal areas of China, taking Zhejiang Province as an example since 1949. They proposed and discussed the spatial distribution dynamics and diffusion processes of religious institutions, using GIS (Zhao et al. 2017). Additionally, Marcus Bingenheimer used GIS to visualize the pilgrimage routes recorded in the 19th century book *Knowing the Paths of Pilgrimage* (*Canxue zhijin* 參學知津) written by Ruhai Xiancheng (如海顯承), showing the pilgrimage network of temples (Bingenheimer 2016).

## 1. Comments on the Studies of Goossaert and Chang

According to my understanding, the most notable studies on the number and distribution of ordination certificates during the early Qianlong period are those conducted by the French scholar Vincent Goossaert and the Chinese scholar Chang Jianhua (常建華). Goossaert's "Counting the Monks: The 1736–1739 Census of the Chinese Clergy"[5] and Chang's "On the Administration of Monks in the Early Qianlong Period" (*Qingshi luncong* 2002), have provided valuable insights into this topic with abundant resources.

*1.1. The Research of Goossaert*

Goossaert discovered two incomplete annual reports (Huangce, 黃册) by the Board of Rites (禮部) to the throne, which were located in the Number One Historical Archives (第一歷史檔案館). These reports covered the periods Qianlong 2.10.1 to Qianlong 3.10.29 (22 November 1737 to 9 December 1738) and Qianlong 4.10.30 (29 December 1738 to 30 November 1739), respectively. They recorded the number of monks and nuns reported by some counties in Shuntian (順天府), Zhili (直隸, now Hebei Province), Shengjing, Shandong, Anhui, Jiangsu, Zhejiang, Hunan, Sichuan, Fujian, Guangdong and Guizhou provinces at that time; However, there were no records or too much missing data to file in Shaanxi, Hubei, Jiangxi, Yunnan, Henan, Shanxi, Gansu, Guangxi, and other provinces in these provinces. Nevertheless, Goossaert found the total number of monks and nuns reported by Shaanxi, Hubei, Jiangxi, Yunnan and other provinces in *Daqing huidian shili* (大清會典實例), so that only Henan, Shanxi, Gansu and Guangxi are completely unknown. Based on the data provided in these two official reports, together with those in *Daqing huidian shili*, Goossaert extrapolated the amount and regional distribution of monks during that period as follows (Table 1):

**Table 1.** Professor Goossaert's estimated numbers of monks and Taoists in each province in 1737 and 1738.

| | Documented Counties | Total Counties | Total from Two Extant Huangce | Extrapolated Total | Taoists to Clergy % |
|---|---|---|---|---|---|
| 1. Documented provinces | | | | | |
| Shuntian | 11 | 25 | 3656 | 8309 | 13.4% |
| Zhili | 94 | 119 | 10,950 | 13,862 | 20.8% |
| Shengjing | 22 | 23 | 4034 | 4217 | 21.1% |
| Shandong | 46 | 107 | 13,284 | 27,469 | 35.3% |
| Anhui | 42 | 56 | 22,481 | 25,576 | 7.0% |
| Jiangsu | 20 | 66 | 9334 | 28,030 | 15.7% |
| Zhejiang | 33 | 78 | 19,886 | 39,428 | 4.0% |
| Hunan | 34 | 69 | 9279 | 11,426 | 16.0% |
| Sichuan | 37 | 125 | 2839 | 9591 | 9.3% |
| Fujian | 58 | 65 | 11,443 | 12,824 | 4.2% |
| Guangdong | 54 | 88 | 12,525 | 20,411 | 7.8% |
| Guizhou | 51 | 60 | 1708 | 2009 | 0.5% |
| 2. Provinces for which the total is known | | | | | |
| Shaanxi | 0 | 79 | | 7911 | |
| Hubei | 0 | 69 | | 29,152 | |
| Jiangxi | 0 | 78 | | 31,099 | |
| Yunnan | 0 | 81 | | 3750 | |
| Total | | | | 275,065 | |
| Remaining | | | | 65,047 | |
| 3. Completely undocumented provinces | | | | | |
| Henan | 0 | 106 | | 26,318 | |
| Shanxi | 5 | 96 | | 19,479 | |
| Gansu | 0 | 57 | | 11,696 | |
| Guangxi | 23 | 98 | | 7554 | |
| Grand Total | 531 | 1545 | | 340,112 (340,111 in fact) | 13.4% |

The main task of Goossaert was to divide the total of 340,112 ordination certificates into different portions and distribute them to each province. According to my understanding, the provinces can be categorized into three groups: (A) those about which he knew the accurate reported amount of ordination certificates at that time; (B) those with incomplete data, only a few from some counties; and (C) those with little detailed information. Shandong (27,469), Anhui (25,576), Hunan (11,426), Shaanxi (7911), Hubei (29,152), Jiangxi (31,099) and Yunnan (3750) were all in Group A. Group B included Shuntian, Zhili, Shengjing, Jiangsu, Zhejiang, Sichuan, Fujian, Guangdong, and Guizhou. And Group C included Henan, Shanxi, Gansu and Guangxi.

For the last two groups, Goossaert calculated the total number of ordination certificates in each province by extrapolation. For group B, he extrapolated using the following formula:

Extrapolated total = total from two extant Huangce/documented counties ∗ total counties

For group C, he deducted the total of the first two groups (275,065) from the grand total (340,112). The remainder (65,047) is the total amount in the relevant provinces. Then, Goossaert portioned out this total of 65,047 into the four provinces according to their population ratios at that time: Henan (12,847,909), Shanxi (9,509,266), Gansu (5,709,526), Guangxi (3,687,725). As a result, the numbers of ordination certificates he obtained for the above four provinces were 26,318, 19,479, 11,696, and 7554 respectively.

By adding the numbers of ordination certificates in each province, we can get 340,111. In this way, Goossaert almost accomplished his proposed task. His analysis and extrapolation were impressive. However, we should also be aware of the following two points:

The extrapolation of the total number of ordination certificates in a certain province based on the rate of documented and undocumented counties, may result in inaccuracies due to the differences among counties.

The distribution of 65,047 ordination certificates to the four provinces in the last group according to the population ratio may probably lead to even greater inaccuracy. This is because the extrapolated number of 65,047, as the remainder based on the extrapolation of Group B, may itself be inaccurate itself, as explained ahead. Moreover, the distribution of monks in these four provinces, including the Central Plains and remote frontiers, would not be so even because of different topographical features. In particular, the extrapolated number of monks—11,696 in Gansu and 7554 in Guangxi—seems to be huge. Even when compared with the figure of 1047 shown in the Huangce that Goossaert mentioned, the number of ordination certificates in Guangxi still seems to have been exaggerated.

### 1.2. The Research of Chang

In addition to common historical resources, Chang's research involved in *The written official reports stored in the palace during Qianlong's reign*, edited by the Palace Museum in Taipei. However, Huangce, the main resource used by Goossaert, was not involved in Chang's research. The distinguishing feature of Chang's research is the statistics on the number of monks based on the data revealed in 37 official written reports from different provinces, as shown in the Table 2 below.[6]

Having issued 340,112 ordination certificates from the years 1736 to 1739, the policy of the central government is gradually reduced the number of certificates. In accordance with this policy, local officials reported fewer ordination certificates each year. As a result, the later the certificate, the fewer certificates issued, and the greater the gap between the existing and the original number of certificates.

**Table 2.** Professor Chang's estimated numbers of monks and Taoists in each province from 1750 to 1753.

| Area | Original[7] | 1750 | 1751 | 1752 | 1753 |
|---|---|---|---|---|---|
| Zhili | | | 9205 | 9151 | |
| Jiangsu | 24,687 (issued in 1749) | 24,299 | 23,981 | | |
| Anhui | 25,576 | | 20,250 | 19,928 | |
| Shandong | 27,469 | 19,876 | 19,489 | 19,004 | |
| Shaanxi | 7911 | | 5491 | 5343 | 5284 |
| Gansu | | 1119 | 1088 | 1075 | |
| Zhejiang | 52,566 | | | 40,300 | 39,926 |
| Jiangxi | 31,099 (including those issued afterwards) | | 23,450 | 23,168 | 22,857 |
| Hubei | 29,152 | | 21,312 | 21,013 | 20,861 |
| Hunan | 11,426 | | 9146 | 8971 | 8861 |
| Sichuan | | | | 7006 | 6933 |
| Fujian | | | | 7305 | 7147 |
| Guangdong | | 10,310 | 10,177 | 9904 | 9782 |
| Guangxi | | 640 | 629 | 616 | 608 |
| Yunnan | 3750 | 2495 + ? | 2413 + 2 | 2338 + 11 | 2298 + 9 |
| Guizhou | | | 1172 | 1158 | 1145 |
| Total | | | | 176,291 | |

Notes: ? means unclear.

### 1.3. Comparison of Goossaert with Chang

In the comparison between Goossaert and Chang, I compared the number of certificates between 1736 and 1739 that Chang found in the official reports with the number extrapolated by Goossaert, as follows (Table 3):

**Table 3.** A comparison of the similarities and differences in the estimated numbers of monks and Taoists in each province in the mid-18th century by Professors Goossaert and Chang.

| Area | Goossaert's Extrapolation on the Number of Ordination Certificate during 1736–1739 | Chang | |
|---|---|---|---|
| | | Number of Ordination Certificate Closest to 1736–1739 | Time of Report |
| Shuntian | 8309[8] | | |
| Zhili | 13,862 | 9205 | 1751 |
| Shengjing | 4217 | | |
| Shandong | 27,469 | 27,469 | Original |
| Anhui | 25,576 | 25,576 | Original |
| Jiangsu | 28,030 | 24,687 | 1749 |
| Zhejiang | 39,428 | 52,566 | Original |
| Hunan | 11,426 | 11,426 | Original |
| Sichuan | 9591 | 7006 | 1752 |

Table 3. *Cont.*

| Area | Goossaert's Extrapolation on the Number of Ordination Certificate during 1736–1739 | Chang | |
| | | Number of Ordination Certificate Closest to 1736–1739 | Time of Report |
|---|---|---|---|
| Fujian | 12,824 | 7305 | 1752 |
| Guangdong | 20,411 | 10,310 | 1750 |
| Guizhou | 2009 | 1172 | 1751 |
| Shaanxi | 7911 | 7911 | Original |
| Hubei | 29,152 | 29,152 | Original |
| Jiangxi | 31,099 | 31,099 | Original (including those issued in 1742) |
| Yunnan | 3750 | 3750 | Original |
| Henan | 26,318 | | |
| Shanxi | 19,479 | | |
| Gansu | 11,696 | 1119 | 1750 |
| Guangxi | 7554 | 640 | 1750 |
| Grand Total | 340,112 | | |

From the above table, we can see that the number of certificates from 1749 to 1752 that Chang found in all provinces is less than the number extrapolated by Goossaert from 1736 to 1739. From this point of view, Goossaert's extrapolation is generally reliable. However, we may also notice the case of Guangdong province: the gap in figures between the two groups is much too great, with the extrapolated number almost twice the number recorded in the official reports, which is likely closer to reality. This may imply that the figure for Guangdong province is exaggerated. Furthermore, in the case of Zhejiang province, the extrapolated number of 39,428 appears to be too far from the 52,566 that Chang has found, with a difference of 13,138. According to Goossaert's formula, the total number of ordination certificates in the four provinces of Henan, Shanxi, Gansu, and Guangxi in Group C should be revised from 65,047 to 51,909 (=65,047 − 13,138).

Chang found that totals for Gansu and Guangxi in 1750 were 1119 and 640 respectively, while Goossaert found the total for Guangxi in Huangce was 1047. Though there are differences, we can assume that the total number of ordination certificates issued in both Gansu and Guangxi between 1736 and 1739, would not exceed 3000. Additionally, the total for Henan and Shanxi during that period was around 49,000. Based on the close location of Henan and Shanxi, adjacent to each other and both in Northern China, we could assume that the densities of monks in these two provinces are equal. Then, if we divide the number of ordination certificates in Henan and Shanxi based on the population ratio, the number for Henan would be 28,000 and 21,000 for Shanxi. This result is quite close to that extrapolated by Goossaert.

Based on the data that Chang found in the official reports, especially the originally issued number of ordination certificates in Zhejiang province, we can conclude that Goossaert's extrapolations were generally reliable, and some amendments should be made towards Zhejiang, Gansu, and Guangxi.

The following part will continue to discuss the amount and regional distribution of Chinese monks after the mid-Qing Dynasty.

## 2. The Number and Regional Distribution of Chinese Monks after Mid-Qing Dynasty

After 1739, Emperor Qianlong aimed to limit the number of monks through strict control over newly issued ordination certificates, rather than abolishing the Certificate altogether. Qianlong's policy involved no longer issuing new ordination certificates, in the

hope that the already issued Certificates could be passed down from one generation of ordained monks to the next. Each monk, whether Buddhist or Taoist, could only recruit one disciple, and the ordination certificate would be passed to the disciple after the master's death, rather than issuing a new Certificate to them. As a result, one ordination certificate effectively covered two monks, the master and their disciple. While local Buddhists and Taoists were theoretically not allowed to adopt disciples, in reality, this was not the case. The government at that time estimated that around 300,000 ordination certificates had been issued by the Board of Rites, and only one disciple was allowed among those monks who owned the Certificate. According to this estimate, the total number of monks (including both masters and disciples) was around 600,000 (*Qing Gaozong shilu* (清高宗實錄) V94).

Based on the above reasons, it is believed that there were at least 680,224 (=340,112 ∗ 2) monks in China between 1736 and 1739, given the 340,112 ordination certificates that were issued during that period. Based on the research of both Goossaert and Chang, the following conclusion can be drawn (Table 4):

**Table 4.** The author's estimation of the actual number of monks in each province in the mid-18th century.

| Rank | Area | Extrapolation of the Ordination Certificate during 1736–1739 |
|---|---|---|
| 1 | Zhejiang | 105,132– |
| 2 | Jiangxi | 62,198[9] |
| 3 | Hubei | 58,304 |
| 4 | Jiangsu | 56,060 |
| 5 | Henan | ?56,000 |
| 6 | Shandong | 54,938 |
| 7 | Anhui | 51,152 |
| 8 | Shanxi | ?42,000 |
| 9 | Guangdong | 40,822 |
| 10 | Zhili | 27,724 |
| 11 | Fujian | 25,648 |
| 12 | Hunan | 22,852 |
| 13 | Sichuan | 19,182 |
| 14 | Shuntian | 16,618 |
| 15 | Shaanxi | 15,822 |
| 16 | Shengjing | 8434 |
| 17 | Yunnan | 7500 |
| 18 | Guizhou | 4018 |
| 19 | Gansu | ?3600 |
| 20 | Guangxi | ?2220 |
| | Total | 680,224 |

Notes: ? means unclear.

(The number for Jiangxi in the above table should be slightly lower, but since it is only a rough estimate, so the estimated number of renewal certificates for the year 1742 will not be revised. Additionally, the estimate for Guangdong Province may be slightly inflated).

From the table above, we can categorize the regional distribution of monks in the mid-Qing Dynasty as follows:

(1) Zhejiang Province has over 100,000 people; (2) Provinces such as Jiangsu, Jiangxi, Hubei, Anhui, Shandong, and Henan have over 50,000 people; (3) Provinces such as Guangdong, Shanxi, Zhili, Hunan, Fujian and Sichuan have between 20,000 to below 50,000 people; (4) Provinces such as Shaanxi and Shuntian have around 15,000 people; and (5) Provinces such as Shengjing, Yunnan, and Guizhou have only a few thousand people.

It is revealed that the Yangtze River's southern area (the middle and lower reaches of the Yangtze River: Jiangsu and Zhejiang) is the center of Buddhism, from which it radiates westward (the middle reaches of the Yangtze River: Hunan, Hubei and Sichuan, etc.), northward (North China: Anhui, Henan, Shanxi, Shandong, Zhili, etc.) and southward (Jiangxi, Fujian, etc.); Meanwhile, southwest China (Yunnan, Guizhou, Guangxi and other provinces), northwest China (Shaanxi, Gansu), and northeast China (Shengjing), which are farther away from the southern area of the Yangtze River, have fewer monks or nuns.

Regarding the proportion of monks and nuns, Mr. Chang Jianhua has only sorted out a few data points for reference. Only the proportion of monks and nuns in Jiangsu Province from 1749 to 1751 and the proportion of monks and nuns originally issued by Hunan Province from 1736 to 1739 can be calculated (Table 5):

**Table 5.** The proportion of monks and Taoists in Jiangsu and Hunan provinces in the mid-18th century.

| Number of Monks | Area |
|---|---|
| More than 100,000 | Zhejiang |
| Between 50,000 and 100,000 | Jiangsu, Jiangxi, Hubei, Anhui, Shandong, Henan |
| Between 20,000 and 500,000 | Guangdong, Shanxi, Zhili, Hunan, Fujian, Sichuan |
| Around 15,000 | Shaanxi, Shuntian |
| Less than 10,000 | Shengjing, Yunnan, Guizhou, Gansu, Guangxi |

In this sense, the religious center of China is situated in the Jiangnan Delta, along the middle and lower reaches of the Yangtze River, especially in Jiangsu and Zhejiang. From this religious center, the influence of religion radiated westward to Hunan, Hubei, and Sichuan, along the middle reaches of the Yangtze River; northward to Anhui, Henan, Shanxi, Shandong and Zhili, in northern China; and southward to Jiangxi and Fujian. Meanwhile, in the areas far from the center, such as the southwestern part (including Yunnan, Guizhou, and Guangxi), the northwestern part (including Shaanxi and Gansu), and the northeastern part (Shengjing), the number of monks decreased significantly.

Regarding the ratio of monks to clergy, Chang's data provides only limited information. Only the ratio of monks to clergy in Jiangsu between 1749 and 1751 and that in Hunan between 1736 and 1739 can be determined (Table 6).

**Table 6.** The proportion of monks and Taoists with ordination certificates in Zhejiang, Shanxi, and Gansu provinces in the mid-18th century.

| Area | Year | Buddhists | Taoists | Taoists to Clergy % (by Chang) | Taoists to Clergy % (by Goossaert) |
|---|---|---|---|---|---|
| Jiangsu | 1749 | 20,674 | 4013 | 19.4% | 15.7% |
| | 1750 | 20,353 | 3946 | 19.4% | |
| | 1751 | 20,069 | 3912 | 19.5% | |
| Hunan | Original (1736–1739) | 9603 | 1823 | 19.0% | 16.0% |

Additionally, they are valuable in figuring out the variations in both the number of monks and the number of ordination certificates in Zhejiang, Shanxi, and Gansu (Table 7).

**Table 7.** The distribution of Buddhist temples in each province in the second half of the 18th century.

| Area | Year | Buddhists | Taoists | Taoists to Clergy % (by Chang) | Taoists to Clergy % (by Goossaert) |
|---|---|---|---|---|---|
| Zhejiang | 1751 (number of newly issued certificates) | 2575 | 183 | 7.1% | 4.0% |
| | 1751 (number of monks without certificates) | 6177 | 287 | 4.6% | |
| | 1752 (number of newly issued certificates) | 2804 | 196 | 7.0% | |
| | 1752 (number of monks without certificates) | 6082 | 274 | 4.5% | |
| Shanxi | 1751 (reduced number of certificates) | 277 | 38 | 16.7% | |
| | 1752 (reduced number of certificates) | 226 | 46 | 20.4% | |
| Gansu | 1751 (reduced number of certificates) | 26 | 8 | 30.8% | |

Goossaert believed that ratio of Taoist monks to clergy was around 13.4% during that period, which may be accurate. In other words, there were at least 600,000 Buddhist monks and 100,000 Taoist monks during Qianlong's reign.

In the following section, my discussion will delve deeper into the regional distribution of Buddhist monks, taking into consideration the number and regional distribution of Buddhist temples during Qianlong's reign. The reasons for leaving Taoist monks aside is twofold: first, Buddhist monks outnumbered Taoist monks; and second, many Taoist monks did not live in temples, making the relationship between Taoist monks and temples was looser than that between Buddhist monks and temples. Additionally, resources on Taoist monks are limited, so further extrapolation and comparison on them may result in inaccuracy, if not outright error.

Between 1764 and 1784, the *Daqing yitong Gazetteer* (大清一統志) was compiled and edited. According to the statistics compiled by the Japanese scholar Kanayama Shōkō (金山正好), this series of books recorded 2396 Buddhist temples at that time. Based on his data, I categorized them based on province as follows (Table 8):

**Table 8.** The distribution and gender ratio of monks in each province in the first half of the 20th century.

| Rank | Area | Average Number of Temples in Each Prefecture | Total Number of Temples in the Province |
|---|---|---|---|
| 1 | Jiangsu | 18.6 | 205 |
| 2 | Zhili | 16.3 | 294 |
| 3 | Hubei | 15.9 | 143 |
| 4 | Zhejiang | 13.7 | 151 |
| 5 | Henan | 13.5 | 175 |
| 6 | Shanxi | 11.4 | 228 |
| 7 | Shaanxi | 10.7 | 128 |
| 8 | Guizhou | 10.2 | 122 |
| 9 | Jiangxi | 9.9 | 139 |

**Table 8.** *Cont.*

| Rank | Area | Average Number of Temples in Each Prefecture | Total Number of Temples in the Province |
|------|------|------|------|
| 10 | Hunan | 9.2 | 119 |
| 11 | Shandong | 8.9 | 107 |
| 12 | Anhui | 8.2 | 106 |
| 13 | Fujian | 6.8 | 82 |
| 14 | Guangdong | 6.0 | 72 |
| 15 | Yunnan | 6.0 | 121 |
| 16 | Sichuan | 5.5 | 122 |
| 17 | Shengjing | 5.0 | 26 |
| 18 | Guangxi | 3.6 | 36 |
| 19 | Gansu | 2.5 | 20 |

From the table above, the provinces with an average of more than 15 included Jiangsu, Zhili, and Hubei. Those with an average between 11 and 14 were Zhejiang, Henan and Shanxi. Provinces with an average between 8 and 11 included Shaanxi, Guizhou, Jiangxi, Hunan, Shandong, and Anhui. Those between 5 and 7 included Fujian, Guangdong, Yunnan, and Sichuan. Provinces with average below 5 included Shengjing, Guangxi and Gansu. Therefore, during Qianlong's reign, officially authorized Buddhist temples were highly centralized along and around the downstream of the Yangtze River, such as Jiangsu, Hubei, Zhejiang, Jiangxi, Hunan, and Anhui. They were also found in northern China, such as Zhili, Henan, Shaanxi, and Shanxi, as well as coastal areas in the southeast, like Fujian and Guangdong, and the southwestern part, like Guizhou, Yunnan, and Sichuan. However, there were even fewer temples in northeastern, Gansu and Guangxi.

The conclusion drawn from the *Gazetteer* about the regional distribution of Buddhist temples is quite similar to that of monks between 1736 and 1739 that we previously analyzed previously, particularly the overall layout. That is to say, the regions along the midstream and downstream of the Yangtze River ranked first in both the number of officially authorized Buddhist monks and temples. From this point of view, it was the center of Chinese Buddhism. The southwestern and southeastern regions came in second and third, respectively. The influence of Buddhism became weaker in the northwestern regions and Guangxi. However, there were some divergences. For instance, the number of Buddhist temples in Northern China ranked high, but that was not the case for the number of Buddhist monks there. In the southwestern regions, such as Guizhou, the number of temples outnumbered monks. Such divergence might be the result of statistics and the policy of the central government. In the Jiangnan Delta where Buddhism was popular, the central government would limit the number of Buddhist temples, and in order to adapt to this policy, the local officials in these regions might have made false reports about the number of monks. While Buddhism was not as popular in the remote regions, the central government would allot more temple quotas of temples there to balance the influence of Buddhism in different regions. The northern regions might be allotted more quotas than other regions due to historical and geographical factors because they were close to the political capital.

In short, though the ordination certificate and the Buddhist monks and temples discussed here were limited to the officially authorized ones, it is still meaningful and valuable for us to understand the development of Buddhism throughout China during Qianlong's reign. It is clear that during that period, Buddhism flourished most along the midstream and downstream of the Yangtze River. The northern regions ranked second, the coastal areas in the southeast and the southwestern regions were third; while the influence of

Buddhism was even weaker in the northwestern regions, the northeastern regions, and Guangxi province.

## 3. Further Comparison

There has been no nationwide census on monks, whether Buddhist or Taoist, since the system of ordination certificates was completely abolished in the mid-Qianlong period, until the Society of Chinese Buddhism conducted one in the 1930s. According to its data, there were more than 267,000 Buddhist temples and 738,000 Buddhist monks all over China in the 1930s. In the case of Taoism, there was no nationwide data about Taoist monks and temples even during the Republic Period. Goossaert has noted that there were about 50,000 Taoist monks were living in temples in 1949 (Goossaert 2004). Since China had experienced many wars before 1949, and many Taoists did not live in temples, there should have been more Taoists during the Republic Period than in 1949 (about 50,000), but the number may have been similar to the figure during Qianlong's reign (more than 100,000), as I extrapolated in the previous section. Because of the low percentage of Taoist monks and the lack of resources, the focus of the next section is still on Buddhist monks.

With the help of other resources, Holms Welch has sorted the statistical data compiled by the Society of Chinese Buddhism, which was published in the 1936 edition of the Shen-pao Yearbook (Shanghai) ( *Shen-pao Yearbook* 1936). I will compare the number of Buddhist monks during Qianlong's reign with that in the 1930s in different provinces and figure out their trends of change (Welch 1967) (Table 9):

**Table 9.** The distributional changes of the number of monks in each province in China in the past 200 years (from the mid-18th century to the first half of the 20th century).

| Province | Extrapolated Number of Ordination Certificate between 1736 and 1739 | Census of Buddhist Monks in the 1930s | Trend |
|---|---|---|---|
| Zhejiang | 105,132 | 107,700 | |
| Jiangxi | 62,198 | 2640 | Obviously down |
| Hubei | 58,304 | 76,040 | Up |
| Jiangsu | 56,060 | 171,760 (excluding 6200 in Shanghai) | Obviously up |
| Henan | ?56,000 | 2960 | Obviously down |
| Shandong | 54,938 | 4730 (excluding 1490 in Qingdao) | Obviously down |
| Anhui | 51,152 | 29,540 | Obviously down |
| Shanxi | ?42,000 | 16,640 | Obviously down |
| Guangdong | 40,822 | 19,120 | Obviously down |
| Zhili | 27,722 | 2100 (Hebei) | Obviously down |
| Fujian | 25,648 | 33,360 | |
| Hunan | 22,852 | 62,400 | Obviously up |
| Sichuan | 19,182 | 158,610 | Obviously up |
| Shuntian | 16,618 | 1340 (Beijing) | Obviously down |
| Shaanxi | 15,822 | 1010 | Obviously down |

**Table 9.** *Cont.*

| Province | Extrapolated Number of Ordination Certificate between 1736 and 1739 | Census of Buddhist Monks in the 1930s | Trend |
|---|---|---|---|
| Shengjing | 8434 | 1480 (including 770 in Liaoning and 710 in Heilongjiang) | |
| Yunnan | 7500 | 37,180 | Obviously up |
| Guizhou | 4018 | 810 | Down |
| Gansu | ?3600 | 490 | Down |
| Guangxi | ?2218 | 460 | Down |
| Grand Total | 680,224 | 738,200 | Maintaining in a stable level with slight raise |

Notes: ? means unclear.

According to the available resources, the total number of Buddhist monks did not change significantly during the two hundred years from Qianlong's reign to the Republic Period, staying within a range between 600,000 to 700,000. Although the census in the 1930s did not cover Taoist monks, as previously discussed, their numbers may have been similar to those during Qianlong's reign. As a result, the total number of monks (both Buddhist and Taoist) remained relatively constant since the mid-Qing Dynasty, despite significant historical changes in the 19th century, such as population growth, the Taiping Heavenly Kingdom Movement, the promotion of education with temple property, and the warlord conflicts.

The table above also shows that the number of Buddhist monks in Northern China, including Shandong, Henan, Anhui, Shanxi, and Shaanxi, declined significantly from 1742 to 1936, while the number in the regions along the midstream and downstream of the Yangtze River and in southwestern China, including Hunan, Hubei, Sichuan, and Yunnan increased considerably. However, the geographical distribution of Chinese Buddhism remained relatively unchanged. Regions along the midstream and downstream of the Yangtze River continued to be the epicenter of Chinese Buddhism, accounting for more than half of all Buddhist monks in China. Zhejiang province previously had the highest number of Buddhist monks, but after the Taiping Heavenly Kingdom Movement, Jiangsu became the top province, and Zhejiang province's population decreased by half. Nevertheless, the number of Buddhist monks in Zhejiang province did not changed significantly due to the movement.

## 4. Conclusions

Emperor Qianlong did not intend to abolish the ordination certificate but stopped issuing new ones in the hope of reducing the number of monks. However, he later found that this policy was futile in controlling the population of monks, as it only increased the number of monks without an ordination certificate. This led to difficulties in selecting Senglu Si (僧录司) and Daolu Si (道录司) officials, as candidates for these positions were required to have their own ordination certificate. According to a source, " since the emperor stopped issuing ordination certificates, there are few monks who fit the position (of Senglu Si and Daolu Si)" (自牒照停止頒給以來，選充無人). This difficulty prompted the emperor to adapt his policy to the needs of administration. He planned to reissue the ordination certificate in 1771 but changed his mind due to the trouble it would cause, ultimately leading to the complete abolition of the ordination certificate. After this abolishment, the number of monks did not increase rapidly, instead maintaining a stable level until the Republic Period. Regardless of the existence of the ordination certificate, the number of monks has never exceeded one million, and it did not change rapidly after the abolishment of the ordination certificate. Therefore, the system of ordination certificates, which had lasted for

over a thousand years, lost its efficacy during Qianlong's reign and was abolished at the end. From this point of view, the system of ordination certificates is not an effective way to control the number of monks. Attention should be paid to the regional distribution disequilibrium of Chinese monks, with the majority located along the Yangtze River and the sharp reduction in the number of monks in Northern China from the Mid-Qing Dynasty to the Republic Period.

The number of monks did not change much before and after the abolition of the ordination certificate system during the Qianlong period. Even from the middle of the Qing Dynasty until the Republic of China, the number of monks in China remained surprisingly stable. This stability was manifested not only evident in the number of monks but also in their regional distribution. While it is commonly known that Buddhism thrived in the middle and lower reaches of the Yangtze River, academic research must rely on data to substantiate this view, including the distribution of monks and monasteries. This data can also provide a basis for explaining many modern Buddhist events. The further decline in the number of Buddhist monks in the north after the mid-Qing Dynasty can only be explained by the rise of folk secret sects[10] and the Boxer Rebellion (義和團運動), which is also a valuable academic issue. Additionally, North China has more farmers and South China has more tenants. The traditional temple economy in China is better suited to the environment in South China, which may be one of the important reasons why Buddhism is weaker in North China than that in South China. Furthermore, during the late Qing and early Republican eras, Nanjing's position as the political capital and the emergence of Shanghai, initially part of Jiangsu province at this time, had a catalytic influence on Buddhism in Jiangsu.

Nevertheless, this article cannot address all the issues mentioned above. It does, however, reveal the impressive stability of traditional Buddhism in China. Many systems and operational modes in China's traditional society demonstrate astonishing resilience, and they are often effective if they are not severely impacted by external forces.

**Funding:** This research received no external funding.

**Institutional Review Board Statement:** Not applicable.

**Informed Consent Statement:** Not applicable.

**Data Availability Statement:** Not applicable.

**Conflicts of Interest:** The author declares no conflict of interest.

## Notes

1 For example, Shi Dongchu's (釋東初) *Zhongguo fojiao jindaishi* (中國佛教近代史, the Modern History of Buddhism in China, 1974).

2 Scholars' research on the history of modern Buddhism was initially derived from the research field of the history of modern thought, such as Ma Tianxiang's (麻天祥) *Wanqing foxue yu jindai shehui sichao* (晚清佛學與近代社會思潮, Buddhism in the Late Qing Dynasty and Modern Social Thoughts, 1992), Li Xiangping's (李向平) *Jiushi yu jiuxin: zhongguo jindai fojiao fuxing sichao yanjiu* (救世與救心: 中國近代佛教復興思潮研究, Saving the World and Saving the Hearts: A Study on the Revival of Modern Buddhism in China, 1993), He Jianming's (何建明) *Fofa de jingdai tiaoshi* (佛法的近代調適, Modern Adjustment of Buddhism, 1998), and Chen Bing (陳兵) and Deng Zimei's (鄧子美) *Ershi shiji zhongguo fojiao* (二十世紀中國佛教, 20th Century China Buddhism).

3 The concept of "revival", proposed by Holmes Welch, has been a topic of controversy in recent years. In the special issue on "Revisiting the Revival: Holmes Welch and the Study of Buddhism in Twentieth-Century China" in *Studies in Chinese Religions*, vol. 3, no. 3, 2017, pp. 197–300, many scholars have examined modern religious issues in relation to politics and other social forces with the goal of uncovering how religious changes have shaped modern political, economic, and social life in China (Goossaert and Palmer 2012). Others have sought to address important historical facts and possible threads of modern Buddhism's development that were overlooked in the previous "revival" paradigm by using concepts such as "restoration", "adaptation", "reconstruction", and "revitalization" (Kiely and Jessup 2016).

4 To a certain extent, there existed a situation of "monks ruling monks" in the Qing Dynasty, but the system that played a crucial role was not the "bureaucratic" style or the top-down monk official system, but rather the patriarchal clan system.

5 *Late Imperial China*, vol. 21, no. 2(2000), pp. 40–85.

6     Similar to the research materials and methods used by Mr. Chang Jianhua, Mr. Yang Jian also compiled a table showing the number of monks and Taoist priests in all provinces during the Qianlong period. This table is contained in Gongzhongdang qianlongchao zouzhe (宮中檔乾隆朝奏摺, the Memorial of the Qianlong Dynasty in the Palace, Series 1–7, Taipei: National Palace Museum, May-November, 1982). The table is basically the same as the one listed by Mr. Chang Jianhua, with the same number in Zhili, Anhui, Shandong, Shaanxi, Gansu, Jiangxi, Sichuan, Guangdong, Guangxi, Yunnan, and Guizhou. However, Fujian province is not listed, and Shuntian is supplemented with other materials instead. The data of Jiangsu and Hunan provinces listed in Mr. Yang Jian's table differ from those in Mr. Chang Jianhua's table, with 20,674 people, 20,353 people, and 20,069 people in Jiangsu Province in 1785, 1786, and 1787, respectively. There were 9603 people before 1785, and 7600 people, 7444 people, and 7349 people in 1786, 1787, and 1788, respectively, in Hunan Province. The data of Jiangsu and Hunan provinces in Mr. Yang Jian's table are slightly lower than those in Mr. Chang Jianhua's table. In addition, in the data of Zhejiang Province in 1787 and 1788, Mr. Yang Jian's table indicates that "there are 6464 people who have no certificate" and "there are 6356 people who have no certificate". There are two groups of data in Hubei Province in 1786, 1887, 1788, one group is the same as that listed by Mr. Chang Jianhua, and the other group is slightly less, with 21,245, 20,936, and 20,771 people, respectively. I believe that Mr. Yang Jian's table differs slightly from Mr. Chang Jianhua's table, and this slight difference in the original data is caused by the constant "reduction" of the data by the officials of the Qing Dynasty when they counted and reported. Because this kind of "reduction" is usually to meet the requirements of Emperor Qianlong, officials at all levels constantly artificially "reduced" the number. Therefore, in the following analysis, I use Mr. Chang Jianhua's tables, which contain a large number of statistical results.

7     The term "Original" in the table means the number of ordination certificates issued from the year 1736 to 1739.

8     According to Zhang (1988). This figure is also decreasing year by year "artificially", which is smaller than the original data checked by Goossaert. The further analysis below uses the data listed by Goossaert, which are more reliable.

9     This number may be slightly smaller, but since it is estimated roughly, I will not revise it by taking into consideration the number of certificates issued in 1742.

10     Dr. Liu Dianli (劉殿利) from Renmin University of China used four indicators to analyze the development of folk religions in the Qing Dynasty: (1) the frequency of religious conflict events, (2) the frequency of religious conflict events per one million people, (3) the frequency of activities of religious leaders, and (4) the frequency of activities of religious leaders per ten thousand people. She found that Hebei, Shandong, Henan, Sichuan, Hubei, Shanxi, Shaanxi, and other provinces were the most active areas of folk religions, which has a certain corresponding relationship with the decline of institutional religions such as Buddhism and Taoism in North China and other northern regions discussed in this article. See Liu Dianli, *Mingjian zongjiao de jieshoudu: dui qingdai mingjian zongjiao de yige lianghua yanjiu* (民間宗教的接受度：對清代民間宗教的一個量化研究, Acceptance of Folk Religion: A Quantitative Study of Folk Religion in Qing Dynasty) presented at the Summit Forum on Chinese Religion and Society, sponsored by Peking University, and published on 8 October 2008.

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
