# Peer review of "The Number and Regional Distribution of Chinese Monks after the Mid-Qing Dynasty"

_religions, doi:10.3390/rel14030317_

Round 1

Reviewer 1 Report

This article has no clear thesis.  I have no idea what I am looking to follow, or what conclusion should be properly drawn.  Despite, clarity, this needs work.

Author Response

Thank you for your suggestion. I have corrected the grammatical errors in the whole text and adjusted the structure according to your suggestion.

Reviewer 2 Report

This article’s choice and success in addressing an issue of major relevance to the study of modern Chinese Buddhism, namely, whether Chinese Buddhism has seen a revival since the 18th century, is admirable. The number of monastics and Buddhist monasteries in a particular place and period is the most significant indication for gauging the evolution of Chinese Buddhism. The mid-18th century repeal by the Qing dynasty of the ordination certificates, which had been in effect for more than a millennium, was a pivotal moment in the evolution of the modern Buddhist institution. The introduction of ordination certificates was perhaps one of the most significant means by which the government administered monasticism during the whole time of imperial China. It reflected the subjugation of ecclesiastical power to monarchy and brought the monastic community under effective secular rule. The author of this paper examines the causes and implications of the termination of this system during the Qing Dynasty. This abolition demonstrated that the Qing Dynasty’s policy on religious administration had failed, but the statistics on the number and geographic distribution of the final issuance of ordination certificates by the Qing regime provide a reliable basis for assessing the situation of Buddhism during that same time frame.

In addition to carefully selecting significant academic topics, the author handles the data with great care. The author evaluates and analyses previous scholarly research on the statistics of Chinese monks between 1736 and 1739, highlighting discrepancies between Vincent Goossaert and Chang Jianhua’s research approaches and selection of materials. The author also attempts to determine the political reasons why data from the same period and location vary, such as when local officials manipulate statistics to meet the needs of the central government. Accordingly, the authors make significant adjustments and additions to the prior studies. This displays an excellent aptitude for processing the material.

The author investigates the variations in the number of Buddhist monks in different locations of China from a microscopic viewpoint, utilizing meticulous evidentiary comparisons, and properly concludes that the flourishing of Buddhism increasingly focused on the lower Yangzi River delta. Regarding the study of the monks’ statistics in each province, the author also makes tremendously relevant suggestions for further research.

Overall, this paper is jam packed of helpful material, brilliantly covers a number of crucial topics in modern Chinese Buddhism and handles them well.

Nevertheless, I feel it is important to reiterate that English is the primary subject that needs to be improved. I have proposed a number of adjustments to the attached version of the first two pages, however the English still needs much polishing. Otherwise, it will be quite challenging for readers to comprehend and absorb what is being presented.

Suggestions for additions or reconsideration:

1. There are still areas in this article where citations need to be added.

2. the writers are a bit unfamiliar with the format of English translations of Chinese books and articles.

3. It is advised that, in addition to the English translation of the Chinese source material, the original Chinese text be supplied for the reader's reference.

4. Buddhism flourished more in Jiangsu than in Zhejiang during the late Qing and early Republican eras, although the author does not consider the impact of the capital’s relocation to Nanjing upon Buddhism as one of the causes. Nanjing’s position as the political capital and the emergence of Shanghai, a metropolis that was initially a part of Jiangsu province at this time, had a catalytic influence on Buddhism in Jiangsu.

Author Response

Thank you for your suggestion. I have corrected the grammatical errors in the full text, revised the quotation and format according to your suggestion, and supplemented the relevant content of Nanjing as the capital of the Republic of China government.

Round 2

Reviewer 1 Report

The revisions are sufficient for publication.